# Defining the Collapse Point in Colloidal Unimolecular Polymer (CUP) Formation

**DOI:** 10.3390/polym14091909

**Published:** 2022-05-07

**Authors:** Ashish Zore, Peng Geng, Yuwei Zhang, Michael R. Van De Mark

**Affiliations:** Department of Chemistry, Missouri S&T Coatings Institute, Missouri University of Science and Technology, Rolla, MO 65401, USA; aszbnd@umsystem.edu (A.Z.); pgkr4@umsystem.edu (P.G.); yz3g9@umsystem.edu (Y.Z.)

**Keywords:** colloidal unimolecular polymer (CUP), single-chain polymer nanoparticle, chain collapse, vibration viscometer, Hansen parameters

## Abstract

Colloidal unimolecular polymer (CUP) particles were made using polymers with different ratios of hydrophobic and hydrophilic monomers via a self-organization process known as water reduction. The water-reduction process and the collapse of the polymer chain to form a CUP were tracked using viscosity measurements as a function of composition. A vibration viscometer, which allowed for viscosity measurement as the water was being added during the water-reduction process, was utilized. The protocol was optimized and tested for factors such as temperature control, loss of material, measurement stability while stirring, and changes in the solution volume with the addition of water. The resulting viscosity curve provided the composition of Tetrahydrofuran (THF)/water mixture that triggers the collapse of a polymer chain into a particle. Hansen as well as dielectric parameters were related to the polymer composition and percentage *v*/*v* of THF/water mixture at the collapse point. It was observed that the collapse of the polymer chain occurred when the water/THF composition was at a water volume of between 53.8 to 59.3% in the solvent mixture.

## 1. Introduction

Self-organization of amphiphilic polymers in aqueous solution is of high potential importance in a variety of applications such as paints, coatings, drug delivery, electronics, agriculture and personal goods. The conformational behavior of polymer chains in different regional environments can be explained using the Flory–Huggins theory. Flory [1] describes conformations of uncharged polymers in terms of the theta condition, which could be a theta solvent or theta temperature. When a polymer chain is in the theta condition, it behaves as an ideal chain where polymer–polymer interactions are balanced with polymer–solvent interactions, and the radius of gyration is equal to the random walk configuration. Any deviation from the theta condition can cause the radius Rg to change either due to swelling in a good solvent, greater than theta, or collapse in a poor solvent, less than theta. However, these conformational changes are more complex for charged polyelectrolytes in solution [2,3,4,5]. 

Colloidal unimolecular polymers or CUPs are nanoscale, charge-stabilized, single-chain nanoparticles made from a single polymer chain and have a well-balanced number of hydrophobic and hydrophilic units [6]. The hydrophilic units can be anionic or cationic. The polymer chain is collapsed into a particle through a simple process called water reduction. Figure 1 shows the schematics of the water-reduction process to form CUP particles. The water-reduction process begins with the dissolving of the polymer in a low-boiling, water-miscible solvent. The boiling point of the solvent should be less than that of water since the solvent will be stripped off at the end of the reduction process. THF, used in this study, is a good example of a water-miscible solvent that has good solubility for many polymers, and has a low boiling point of 66 C. The next step is to form the salt or ionic group, in this case, by neutralizing the acid groups—carboxylic acids—using any base such as sodium hydroxide, triethyl amine, ammonia, etc. The base should be added slowly, preferably using a peristaltic pump. Due to the low dielectric of THF, the carboxylate anion and the sodium counterion exist as a tight or intimate ion pair. The repulsive force between the carboxylate anions on the polymer chain is negligible. The polymer chains form inter-chain and intra-chain salt association (see Figure 2) via the sodium-carboxylate group, which causes a small rise in viscosity. The next step is to add water very slowly (1.8 g/min) with stirring to ensure homogenous composition of the solvent throughout the mixture. For water, which is a poor solvent for the polymer, any localized spikes in concentration can cause the polymer to precipitate instead of a proper unimolecular collapse. As the water is being added, the dielectric of the solvent mixture increases [7], as seen in Figure 3. The carboxylate anions start repelling each other more strongly over a longer distance as the dielectric of the media increase. As a result, the polymer chain will become more elongated, and the viscosity will increase as more water is added. This trend will continue until the concentration of water in the solvent mixture reaches a point where it becomes a poor solvent for the polymer and the chain collapses into a spheroidal particle. Here, the transition from coil to globule is triggered by changing the dielectric and solubility parameters of the solvent. The changes in the thermodynamic quality of the solvent makes the polymer–polymer interactions stronger than polymer–solvent interactions, which causes the chain to collapse into a globule. The collapse of the chain is such that the hydrophobic segments form the interior of the particle, and the charged groups are on the surface, as shown in Figure 4. The self-organization of polymer chains into CUPs is similar to that of micelle formation in surfactants. CUPs have a lot of utility in the field of coatings due to their zero-VOC (volatile organic content), low cost and easy synthesis. They have been used as a resin that can be cured with melamine or aziridine [8,9], an additive for the freeze–thaw stability and wet-edge retention [10] of latex paints or as a catalyst [11]. They can also be potential drug-delivery systems. CUPs have been extremely useful in studying the properties of bound or surface water [12,13], and in understanding water evaporation behavior [14], the electroviscous effect [15] and surface tension [16]. 

A polyelectrolyte or a polymer chain containing several ionic groups can form many different conformations depending on the charge density on the chain and its solvent environment. The conformations of polyelectrolytes have been modeled using an electrostatic blob and scaling theory, which was first developed by DeGennes and Pfeuty and later reviewed by Dobrynin [17]. According to the theory, a neutral polymer in a poor solvent such as water will collapse into a spheroidal globule. When charges are present on the chain, it collapses into an electrostatic blob, dumbbell or a pearl necklace, depending on the fraction of charges present on the chain. Kirwan [2] observed conformational changes for polyvinyl amine in water at different pH values. At a low pH = 3, the polymer chain was highly charged and in an extended conformation. Increasing the pH transitioned the chain into a pearl necklace structure. Above pH 9, the polymer collapsed to a globule due to attractive hydrophobic interactions between polymer segments in a poor solvent condition. Similar observations were made by DeMelo [3] using polyacrylic acid by going from high pH to low. The conformational behavior in polymers allows the synthesis of polymers capable of forming a single-chain nanoparticle [18,19,20]. The coil-to-globule transition can be triggered by changing the temperature [21] or by changing the solvent quality such as solvent composition [4,22,23], dielectric [24], or pH [25].

Li [26] used hydrophobic blocks of the anticancer drug paclitaxel and grafted it onto blocks of polyether ester to produce a self-assembled multichain polymeric micelle as a drug-delivery system. When the block co-polymer was placed in an aqueous environment with adjusted pH, the hydrophilic polyether ester was oriented into the water phase, leaving the hydrophobic paclitaxel oriented toward the interior domain. Morishima reported micelle-like behavior in single-chain polyelectrolytes [27] using a random copolymer of a 1:1 monomer ratio of hydrophilic and hydrophobic monomers. The chains were collapsed into unimolecular micelles by dissolving the polymer at very low concentration in aqueous NaOH to obtain particles of 5.5 nm in diameter. The use of a change in solvent composition to achieve a coil-to-globule transition such as CUPs was reported by Aseyev [4]. In their work, polymethacryloyl ethyl trimethyl ammonium methyl sulfate (PMETMMS) was examined in a water/acetone mixture wherein acetone was a non-solvent. Collapse of the polymer chains occurred at a 0.80 mass fraction of acetone in aqueous solution, as observed by a decrease in the reduced viscosity, the radius of gyration and hydrodynamic radius. In an earlier report on the synthesis of CUPs [6], viscosity was used to determine the composition of THF/water mixture required for the coil-to-globule transition of MMA–MAA copolymer. Similar to Aseyev’s observation, the viscosity of the solution dropped when the polymer chains collapsed into CUP particles. The transition occurred at roughly 60% water and 40% THF composition for the CUP example. It should be noted that, unlike other studies, in the CUP system, the good solvent is removed after the particle forms. This solvent removal secures the polymer’s spheroidal conformation and removes all VOCs. 

Non-unimolecular collapse can also be observed in the case of polyurethane dispersions, which are used in the coating industry, wherein the polymer is synthesized in acetone and then followed by the addition of water. When the acetone is removed from the resin blend, the chain collapses into multi-chain aggregates/non-unimolecular particles with diameters of approximately 25 nm [28]. For CUPs, the concentration of polymer in the solution is low enough to prevent chain overlapping or entangling, thereby ensuring that the collapse is unimolecular/single-chain. The unimolecular collapse is confirmed by measuring the particle size using dynamic light scattering (DLS) and overlapping its distribution with the particle size, calculated from the absolute molecular weight measurements using GPC. For unimolecular collapse, the measured and calculated size distribution match [6,20]. The addition of the non-solvent water will alter the dielectric and solubility parameters. The changing solvent composition will have an affect, which will be highly dependent upon the polymer composition in terms of ionic groups and the size and number of hydrophobic groups. The point of collapse has both charge effects and solubility considerations.

The process of water reduction leading to the collapse of the polymer chain into a particle can be tracked by measuring the viscosity as water is being added to it. Figure 1 illustrates the conformational changes of a polymer and viscosity behavior during the water-reduction process. The composition of the water/THF mixture where the polymer chain transitions from a coil to globule is called the collapse point or collapse composition. In an earlier publication describing the synthesis of CUPs [6], preliminary work for the determination of collapse composition was conducted using a cone-and-plate viscometer (Brookfield). The method used was a batch process wherein the water-reduction process was carried out in a separate vessel, and small aliquots of the sample were taken from that solution at regular intervals for viscosity measurement. As the measured sample was not returned to the solution, the concentration had to be corrected every time. Additionally, due to the high volatility of THF, it was difficult to prevent evaporation loss despite the enclosure provided by the instrument. The amount of sample required to measure the viscosity on the instrument was very small (0.5 mL). Hence, even a small loss in THF would significantly change the composition of the solvent. 

Due to the complexity, tediousness and high error margin of the cone-and-plate system, a new protocol was developed to measure viscosity continuously during the reduction using a different type of viscometer called the vibration viscometer. The set-up developed in this study allowed, for the first time, rapid and precise determination of the collapse point where the polymer chain transforms into a collapsed particle. This further enabled us to study the effects of polymer chain composition on the solvent composition required for collapse. The structure of the polymer was also changed by using different amounts and sizes of hydrophobic monomers, and the type of base used for neutralizing the polymer was also investigated. The technique developed in this study can potentially be used to study Flory–Huggins collapse behavior in other polymeric systems. 

## 2. Materials and Methods

### 2.1. Materials and Synthesis 

The method used for purification of materials, synthesis of polymers and reduction process to form CUP particles is reported elsewhere [12]. The methodology and procedure for characterization of polymer (molecular weight, acid number and dry polymer density) and particle size measurement (DLS) of CUP particles is also reported in our previous work [12]. The molar quantities of monomers—methyl methacrylate (MMA), butyl methacrylate (BMA), ethyl methacrylate (EMA), methacrylic acid (MAA), initiator (AIBN) and chain transfer agent (1-dodecanethiol)—used for the synthesis of polymers made for this study are mentioned in Table 1. BMA and EMA were purified by passing them through a basic alumina column. EMA was further purified by distilling it at atmospheric pressure, while BMA was distilled under reduced pressure. AIBN and 1-dodecanethiol were purchased from Sigma-Aldrich. 

### 2.2. AFM Imaging

The AFM images were obtained using the Bruker Dimension Icon instrument, Bruker, Billerica, MA, USA. For preparing the dry samples, 5 μL of 0.0002% Polymer 2 CUP solution was deposited onto freshly cleaved muscovite mica (Ted Pella, Inc., Redding, CA, USA) and air dried for 3 min. The final traces of water were removed by drying with compressed air. Atomic force imaging was conducted by utilizing ScanAsyst mode in air, with ultrasharp 14 series (NSC 14) tips purchased from NANOANDMORE.

### 2.3. Viscosity Measurements

Viscosity was measured using a tuning fork vibration viscometer SV 10A from A&D company Ltd., Tokyo, Japan. with an accuracy of 3% and repeatability of 1%. The instrument has two sensor plates in a tuning fork arrangement that vibrate at a natural (resonant) frequency of 30 Hz inside the sample fluid. Viscosity is then calculated based the amount of electric current required to drive and maintain the sensor plates at a constant vibration amplitude against the viscous resistance of the sample fluid. The instrument can record viscosity with time by using computer software called RsVisco. The viscosity measurement for the determination of collapse point can be conducted using a batch process and continuous process. However, both methods were first tested for factors such as temperature stability, loss of material, etc. 

#### 2.3.1. Testing the Batch Process for Loss of Solution

The batch process was tested using 60 mL of THF/water mixture (75/25 volume ratio) in a screw-top container covered with a lid. The sample was allowed to equilibrate to ambient temperature, which was between 22.5 ± 0.5 °C, before making measurements. The tuning forks were dipped into the sample for 10s to make each measurement. When the sample was removed from the tuning fork, a small amount of solution remained on them. The tuning forks were then immediately washed with DI water to remove the retained solution. The measurement and cleaning were repeated 35 times and the sample was weighed at the end to determine the loss of sample during the entire process. The experiment was repeated three times to obtain the average loss of sample.

#### 2.3.2. Testing the Continuous Process for Collapse-Point Determination

The continuous process was tested using 75/25 volume ratio of THF/water mixture. The container used for measurement was a 120 mL capacity polypropylene beaker with a lid. The lid had a slit big enough for inserting the tuning fork and temperature probe into the beaker, and a small hole for inserting the tubing that would deliver water. The lid helped to reduce the evaporation of THF from the solution. Approximately 60 g (exact amount should be known for concentration calculations) of stock solution was transferred into the container, and a magnetic stirring bar was added and closed with the lid. The experimental set-up is as shown in Figure 5. The beaker was placed in a water bath to keep the temperature constant. The CUP was then placed on a stirring plate with a piece of polystyrene foam placed, to insulate the beaker from the heat of the stirring plate. Stirring was then initiated making sure that the stirring bar stayed stable at a constant rotation speed. The stability of the viscosity reading was checked by measuring the base solution for a few seconds while stirring. When the readings stayed stable to ±0.2 cps, the set-up was ready for the addition of water. Water was added into the solution using a peristaltic pump at a rate of 1.8 g/min. The tip of the pump tubing, when inserted into the beaker, was kept close to the stirring bar to ensure quick mixing of water into the solution. Viscosity measurement was initiated, with viscosity and solution temperature being recorded at fixed regular time intervals of 20 s during the water-addition process using RsVisco software Ver.1.13V, A&D company Ltd, Tokyo, Japan. Based on the flow rate of the pump and time, the amount of water added was calculated, which gave the % volume of water present in the THF/water solvent composition. A plot of viscosity against % volume of water in the THF/water solvent composition was used for analysis.

#### 2.3.3. Collapse-Point Determination of CUP Polymer Using Continuous Process

The experimental procedure used for collapse-point determination of CUP polymer using a continuous process was identical to the batch process, except for the use of a polymer stock solution instead of THF/water mixture (75/25 *v*/*v*). The polymer stock solution was prepared by dissolving the polymer in THF to make a 15% *w*/*w* solution. Then, 1 M sodium hydroxide solution was added in an amount such that all the acid groups were neutralized, and the pH was 8.5–9. Finally, pH-adjusted (pH = 8.5) deionized water was slowly added using a peristaltic pump at a rate of 1.8 g/min. The amount water added was such that the final solvent composition in the polymer stock solution was 25% water and 75% THF. The addition of water increased the temperature of the mixture; hence, before making viscosity measurements, the polymer stock solutions were equilibrated to 22.5 ± 0.5 °C for all the polymers measured. After temperature equilibration, the viscometer measurements were initiated, and the remainder of the water was added at 1.8 g/min. The addition of water was continued for 35 min before ending the experiment. A plot of viscosity against % volume of water in the THF/water solvent composition was made to determine the collapse point or collapse composition. 

## 3. Results and Discussion

### 3.1. Polymer Synthesis and Characterization

To understand the effect of polymer composition, the polymers used in this study were made with different monomer ratios of hydrophobic (MMA, BMA, EMA) and hydrophilic (MAA) monomers. The use of BMA and EMA provides a larger size of hydrophobic group as compared to MMA. Table 2 shows the acid number, density and molecular weight of the copolymers used for this study. The molecular weight and density of the dry polymers are required for calculating the particle size. 

### 3.2. Particle Size Analysis

Table 3 shows the measured particle size for the copolymers and calculated particle size from the absolute number average molecular weight. The diameter of the CUP particles was calculated from its molecular weight using Equation (1).
(1)$$d=\sqrt[{3}]{\frac{6M_{w}}{\pi N_{A}\rho_{p}}}$$
where *d* is the diameter of the particle, *M_W_* is the number average molecular weight of the CUPs, *N_A_* is Avagadro’s number and *ρ_p_* is the density of the dry polymer. As seen from the results, the diameter of the CUP particle increases with an increase in molecular weight. These results are consistent with our previous work and observations made with globular proteins [29,30]. For a unimolecular collapse into a sphere, the particle size measured from DLS should be close to the particle size calculated from the molecular weight using Equation (1), as shown into Table 3. The data show excellent agreement between GPC and DLS diameters.

### 3.3. AFM Image of CUP Particles

Imaging of an isolated single CUP particle is difficult due the formation of particle cluster or aggregation when drying the CUP solution. Figure 6 shows a dense aggregation of Polymer 2 CUP particles, whereas Figure 7 shows sparsely aggregated Polymer 2 CUP particles. Based on the height analysis, the diameter of the particles was found to range from 4.8 to 5.9 nm. This is closer to the average diameter of 5.38 for Polymer 2 measured on a DLS instrument.

### 3.4. Charge Density of the CUP Particle

The charge density *ρ_v_*—i.e., the number of charges per unit area (nm^2^) on the CUP surface—can be calculated using equation 2, where n and m are the number of hydrophobic monomers 1 and 2 for each unit of hydrophilic monomer in a repeat unit, and are also mentioned as a monomer ratio (e.g., n:1 of MMA:MAA) in Table 3; *M_W_* is the molecular weight of the CUP; *M_H_*_1_ and *M_H_*_2_ are the molecular weights of hydrophobic monomers 1 and 2; *M_i_* is the molecular weight of the hydrophilic monomer; and *r* is the radius of the CUP particle.
(2)$$\rho_{v}=\frac{M_{W}}{4\pi r^{2}\left(n\times M_{H1}+m\times M_{H2}+\ldots +M_{i}\right)}$$

### 3.5. Collapse-Point Determination Using Vibration Viscometer

Unlike a cone and plate viscometer, a vibration viscometer can be used to measure viscosity without removing a sample, and can measure the viscosity with active stirring. The water-reduction process and viscosity measurement can be conducted in the same vessel, either through a batch- or continuous-measurement process. 

#### 3.5.1. Batch Process for Collapse-Point Determination Using Vibration Viscometer

Prior to using the batch process for measuring the collapse point of polymers, it was tested for stability of temperature, loss of material, etc. The measurement temperature in the batch process was easily controlled by allowing the sample to equilibrate to the required temperature before making measurements. However, the average loss of material measured by the test experiment was 4.5% by weight after the 35 measurements. The loss of THF due to evaporation can be assumed to be minimum because the sample was enclosed while equilibrating the temperature. Additionally, the measurements on the vibration viscometer were much faster (10 s), which reduced the exposure to air and kept the loss of THF to a minimum. The major contributor to loss of material was the cleaning process of the tuning fork between each measurement. Any sample retained on the tuning fork was lost during the cleaning process. The loss material is very difficult to account for due to inconsistent loss every time and the tedious process of back calculation. This method does have the advantage of giving accurate viscosity values, since each measurement places the vibrating paddles at the correct depth. Due to the loss of material and labor-intensive nature, it was necessary to develop a continuous process for viscosity measurement. 

#### 3.5.2. Continuous Process for Collapse-Point Determination Using Vibration Viscometer

In the continuous process, the viscosity measurements were made as water was being added to the solution. The set-up made for the continuous process for collapse-point determination using the SV 10A vibration viscometer is shown in Figure 5. The set-up was derived experimentally in order to optimize certain parameters involved in the water-reduction process and viscosity measurement. The test experiment performed to derive the set-up and optimize the parameters is described in Section 2.3.2. Figure 8 shows the results of the test experiment, which is the plot of viscosity against the percentage volume of water in the THF/water solvent composition for the water addition to 75/25 *v*/*v* THF/water mixture, and two curves that start with no water. 

Parameter optimization for continuous process:Temperature control and heat of mixing

Temperature is one the key parameters that can change the viscosity of the solution. Hence, for our collapse-point determination experiment that measured the changes in viscosity, the control of temperature was very critical. One of the major sources of heat in the experiment was the heat of mixing, which evolved by adding water to THF. As water was added during the water-reduction experiment, the temperature of the solution rose. Figure 6 shows the temperature profile during the addition of water to THF. The plots were obtained by making three different changes to the experimental set-up, which are described in Section 2.3.2. Plot A is the temperature curve for the addition of water to pure THF, measured on the set-up in Figure 5 without a water bath. Plots B and C are the temperature curves, measured on the set-up in Figure 5, for the addition of water to pure THF and the THF/water mixture of 75/25 *v*/*v* composition, respectively. The difference in the temperatures observed in plots A and B clearly demonstrates the control or mitigation of temperature rise provided by the water bath. It should be noted that the pump added a small amount of heat to the system at a steady rate due to friction, about 0.5 degrees over thirty minutes. The plastic beaker was not a good conductor of heat, and the initial addition of water shows a small, rapid rise due to slow heat transfer regardless of the initial water composition. The temperature profile of the addition of water to pure THF while using a water bath (Plot B) shows a temperature rise of 2.3 °C from 0.0% (22.2 °C) to 43.5% (24.5 °C). The temperature profile (Plot C) of the third method, wherein the water was added to the THF/water mixture of 75/25 *v*/*v* composition while using a water bath, shows a temperature rise of 1.6 °C from 25.0% (22.1 °C) to 54.8% (23.7 °C). The addition of a 25% volume of water followed by equilibrating the temperature before beginning to measure the viscosity further helped in mitigating the rise in temperature by 0.7 °C. Another factor to consider is the stability of the temperature. Plot B shows the temperature reaching a plateau sooner than in Plot C. Plot C shows a stable temperature with a change of less than 0.5 °C from 40% onwards. For a collapse-point measurement for the CUP polymers, which will be discussed in the later section, the critical data required for collapse-point determination are 40% and above, as seen in Figure 9. Therefore, it was concluded that adding a 25% volume of water and using a water bath gave a minimum rise in temperature and a stable temperature with a change of less than 0.5 °C where the critical viscosity data were acquired. Hence, a polymer stock solution, created by adding a 25% volume of water, as described in Section 2.3.3, was prepared. The heat evolved during this stage was easily dissipated by allowing the polymer stock solution to equilibrate to a constant temperature. The steady temperature within 0.5 °C after a 40% water volume mitigated the changes in viscosity caused by temperature variations. Another minor source of heat was from the stirring plate. The temperature of the stirring plate increased with time and was, therefore, higher toward the end of the experiment. The heat transfer from the stirring plate was avoided by placing a piece of insulating material, polystyrene foam, as shown in the experimental set-up (Figure 5), and by allowing the heated stirring plate to cool before beginning the next run. 

2.Loss of material and evaporation of THF

There was material lost during each measurement in the batch process, as described earlier, wherein the tuning fork retained a small amount of the solution. In the continuous process, however, there was no such material loss using this mechanism. The tuning fork was immersed once, and multiple measurements were made with time. Another source of material loss was due to the evaporation of THF, since the container was not completely sealed. To measure the loss of THF through evaporation, the set-up was tested using a THF/water mixture of 75/25 *v*/*v* composition. A 60 g mixture was transferred to the beaker and was stirred for 40 min, which was the typical time for each run, and then weighed again. The loss of mass after 40 min was 1.2%, which was assumed to be due to the evaporation of mainly THF. The loss of material in the continuous process being less than that of the batch process, makes it a better choice for collapse-point determination.

3.Measuring stability while stirring

The major benefit of using a tuning fork vibration viscometer SV 10A is the ability to mix or stir while making viscosity measurements. During the water-reduction process, sufficient stirring is important to make the solution as homogenous as possible and to minimize localized concentration spikes. The vibration viscometer shows very stable viscosity measurement, as seen in Figure 9, with minimum noise or fluctuations in values. The size of the magnetic stirring bar, size of the stirring plate, and RPM determination required some trial-and-error. A simple test to determine the stability of the measurement can be performed by using a sample such as water and observing the fluctuations in the viscosity value. For the set-up used in this work (Figure 5), a small-sized stirring plate (1.8 × 1.8 in., Cimarec^TM^ i micro stirrers by Thermo Fisher Scientific, Waltham, MA, USA) and a stirring bar of 1 cm in length was used. The observed noise in viscosity was about ±0.02 cps on an SV 10A instrument. 

4.Changes in solution volume and real viscosity

Figure 10 shows a diagram of the tuning fork used in the SV 10A instrument. To measure the correct viscosity of a given sample, the tuning forks must be immersed into the solution such that the level is at the curved notch labelled as B (shown in Figure 10). If the solution level is at any other level along the tuning fork, the viscosity value is not correct. In the continuous measurement process, the water was added constantly and, therefore, the level of the solution rose over time. Therefore, the level of the solution could not be maintained at point B while the water was being added. Hence, the viscosity values measured during the water-reduction process were not the correct values. Despite the incorrect viscosity, the overall plot of viscosity against time shows the trend or behavior of viscosity during the water reduction, which is sufficient for determining the collapse point. The result of plotting viscosity against the percentage water volume for a sample of THF/water mixture of 75/25 *v*/*v* composition is shown in Figure 9. Since no polymer was present in the sample, the change in viscosity was due to the increase in the amount of solution causing increasing levels of immersion of the tuning fork. The viscosity shows an increase with time, which is non-linear until point B* and then becomes almost linear. The viscometer is based on the principle of resistance to vibration of the tuning fork inside the sample. As the tuning fork is immersed deeper inside the sample, higher resistance to vibration can be expected due to the increase in the surface area of contact between the sample and the tuning fork. This should result in higher viscosity values as the tuning fork is immersed deeper into the sample, as evident from the results (Figure 9). The changes in the dimensions of the submerged area of the tuning fork cause the non-linear increase in the beginning until point B*. Beyond point B*, the dimensions of the tuning fork remain constant and, hence, the increase in viscosity becomes linear. Knowing the behavior of the viscometer in the absence of any polymer, we can now test the actual polymer solution for its behavior. 

#### 3.5.3. Optimization of the Continuous Process and Measurement of Collapse Point for Polymer Samples

Based on the collapse-point experiment in the previous paper [6], it was certain that the polymer would not collapse at a low concentration of water in the solvent mixture. This is another reason why, prior to making measurements, water was added to the neutralized polymer solution, such that the composition of solvent reached 75/25 *v*/*v* of THF/water in order to create a polymer stock solution. The 25% water added up-front also reduced the total time of the collapse experiment carried out on the water reduction set-up (Figure 5). Reducing the total experiment time also kept the loss of THF via evaporation to a minimum. Moreover, the collapse occurred at a higher concentration of water, so the initial measurements were not critical. The amount (25%) of water added prior to the measurements was also optimized for the small volume of the beaker used for the water reduction set-up. 

The polymer stock solution must be charged into the set-up beaker in a known amount (which should be roughly 60 mL) such that the tuning fork was in the solution to approximately point A (Figure 6). Figure 11 shows the water reduction plot (viscosity against percentage volume of water present in the THF/water solvent composition) for Polymer 2 as it passes through different stages of the tuning fork (Figure 8). The initial level of solution for this run was slightly below the recommended point A, which led to the change in slope at point A. As expected at the beginning, the viscosity values are not linear, which is due to the irregular shape of the tuning fork that begins from point A*. Beyond point B*, the shape of the tuning fork stays regular and consistent. When the solution level crosses point B*, the viscosity increases in a linear trend for a while and then curves, and later starts to drop linearly. The amount of water (25%) added prior to the measurement and the initial solution level at point A are optimized such that the collapse happens roughly around the midpoint between point B* and the full capacity of the beaker. The data in the non-linear region (before point B*) are not critical for our study. The data after point B* are used for collapse-point determination. The increase in the viscosity was due to the polymer chains expanding into a rod-like conformation. It is noteworthy that the viscosity increase occurred even though the concentration of the polymer in the solution was diluted by the addition of water. The drop in viscosity was due to the collapse of the polymer chain into a particle. The broadness of the curvature was due to two opposing effects happening at the same time. There was a decrease in viscosity due to the polymer chains beginning to collapse, but at the same time, the water level increased and immersed the tuning forks deeper, which increased the viscosity value. Later, when the chains collapsed completely, the viscosity started decreasing steadily. The linear rise and linear drop were fit to straight lines and the intersection of the lines was recorded as the collapse point or collapse composition for the given polymer. 

### 3.6. Collaspe-Point Behavior of Different CUP Polymers

The collapse point composition measured for all the polymers used in this study are given in Table 4. Polymers 1 and 2 show the effect of size/molecular weight on the collapse composition. The collapse compositions for both polymers were very similar, indicating that size has no major influence on collapse point, nor does the final charge density, as long as the charges per repeat unit do not change. Polymers 3 and 4 are high and low charge-densities and differ in the amount of hydrophobic and hydrophilic groups present in the polymer chain. Polymer 3 (MMA:MAA::6:1) has more hydrophilic groups and shows collapse at a higher percentage water volume concentration (59.34%) as compared to Polymers 1 and 2 (MMA:MAA::9:1). Polymer 4 (MMA:MAA::18:1) on the other hand has less hydrophilic groups and shows collapse at lower % water volume concentration (54.36%) as compared to Polymers 1 and 2 (MMA:MAA::9:1). These results indicate that hydrophobic and hydrophilic amounts in the polymer chain have a significant influence on the collapse composition. When the polymer chain has more hydrophilicity present, it requires more water in the solvent composition to trigger collapse. Polymers 5 and 6 validate the results observed using different hydrophobic monomers than MMA. As expected, Polymer 5, which has a higher number of hydrophobic units, shows collapse at a lower percentage water volume composition (53.8%). Similarly, Polymer 6, which has fewer of the hydrophobic units, shows collapse at a higher % water volume composition (56.62%). Another way to change the structure of the polymer chain is by using a different base to neutralize the acid groups (hydrophilic groups). Polymer 2 was also neutralized with triethyl amine (Polymer 2*) instead of sodium hydroxide, and then measured for collapse point. The use of triethyl amine shifts the percentage water volume composition (56.24%) to the lower value as compared to the use of sodium hydroxide (57.49%). This shift is because the triethyl amine quaternary counterion is solvated well by organic solvents such as THF. The sodium ion requires more water to solvate, since each sodium ion can be associated with up to six water molecules [31]. The collapse point results from all the polymers indicate that the higher the hydrophobicity of the polymer chain or counter ion, the lower the percentage water volume required to trigger the chain to collapse, and vice versa. This behavior could be due to the differences in the solubility of polymer chain in the water THF mixture for different fractions of hydrophobic and hydrophilic units.

All the CUP polymers in this study collapse at percentage water volumes between 53 to 60%. The dielectric of these solvent mixtures is between 45 to 50. The dielectric obviously plays an important role in determining the collapse point. When the polymer chain collapses, the charges on the chains repel each other strongly to conform the chain into a spheroidal particle. The strong repulsion is also responsible for an even surface-charge distribution, which is required for a stable particle. Hence, a minimum dielectric must be required to separate the ion pairs (acid groups and counter-ion) on the polymer chain, enough that the charges repel each other strongly. Figure 12 shows the dielectric of the collapse concentration against the distance between the charges on the polymer chain. Polymers 3 and 4 have a distance of 1.14 nm and 3.1 nm, respectively, and show a dielectric of 49.6 and 45.9 for the mixture at the collapse point, respectively. Ionic repulsion between the charges is directly proportional to the dielectric, and this is also evident from the viscosity curve in Figure 9, where the chain extends due to ion–ion repulsion as the dielectric increases. Hence, charges separated by a short distance should require lower dielectric as compared to charges separated by a long distance to achieve similar ionic repulsion. If we assume the minimum dielectric required for strong ion–ion repulsion is reached at collapse point, then Polymer 3, where charge separation was shorter, should have required a lower dielectric than Polymer 4, where charge separation was longer. However, Figure 12 shows the opposite trend for all the polymers. Hence, it is very likely that the minimum dielectric required for strong ion–ion repulsion to provide a stable particle must be below the collapse-point dielectric. At the collapse point, the polymer–polymer interactions becoming stronger than polymer–solvent interactions triggers the collapse. 

### 3.7. Comparision Using Hansen Solubility Parameters

Solubility parameters, also known as cohesion energy parameters, are derived from the energy required to convert a liquid to a gas state. The energy of vaporization is a direct measure of the total cohesive energy holding the liquid molecules in the liquid state. Hildebrand and Hansen solubility parameters are the two most widely used measures of solvent–polymer compatibility for determining whether a substance is a good solvent or nonsolvent for a given polymer. Hildebrand is a single parameter, *δ*, defined as the square root of the cohesive energy density: (3)$$\delta ={\left(\frac{E}{V}\right)}^{1/2}$$
where *V* is the molar volume of the pure solvent and *E* is its (measurable) energy of vaporization. The Hansen solubility parameter splits the total cohesive energy *E* into three major intermolecular interactions: (nonpolar) dispersion forces, (polar) permanent dipole–permanent dipole forces, and (polar) hydrogen bonding. The nonpolar cohesive energy (*E_d_*) is derived from induced dipole forces, and is also referred to as atomic or dispersion interactions. The polar cohesion energy (*E_p_*) results from inherent molecular interactions and is found in polar (non-centrosymmetric) molecules. The hydrogen bond cohesive energy (*E_h_*) is the attractive interactions between a hydrogen atom from a molecule or a molecular fragment X–H—in which X is more electronegative than H—and an atom or a group of atoms in the same or a different molecule in which there is evidence of bond formation. The total cohesive energy *E* is the sum of the individual energies that make it up: (4)$$E=E_{d}+E_{p}+E_{h}$$

Dividing the cohesive energies by the molar volume gives the square of the total (or Hildebrand) solubility parameter as the sum of the squares of the Hansen components: (5)$$\frac{E}{V}=\frac{E_{d}}{V}+\frac{E_{p}}{V}+\frac{E_{h}}{V}$$
(6)$$\delta^{2}=\delta_{d}^{2}+\delta_{p}^{2}+\delta_{h}^{2}$$

The Hansen parameters (*δ_d_*, *δ_p_*, *δ_h_*, *δ_T_*) for water and THF are 15.6, 16.0, 42.3, 47.8 and 16.8, 5.7, 8.0, 19.4, respectively [32]. For a mixture of solvents, the Hansen parameters can be calculated based on the volume fractions (*ϕ*_1_ and *ϕ*_2_) of the solvent present in the mix using Equations (6)–(9). The Hansen parameters calculated for solvent composition at the collapse point for Polymers 1–6 are shown in Table 4.
(7)$$\;\delta_{d}=\delta_{d1}\phi_{1}+\delta_{d2}\phi_{2}$$
(8)$$\delta_{p}=\delta_{p1}\phi_{1}+\delta_{p2}\phi_{2}$$
(9)$$\delta_{h}=\delta_{h1}\phi_{1}+\delta_{h2}\phi_{2}$$

The Hansen parameters and the interaction radius (*R*_0_) for the homopolymers shown in Table 5 are experimentally measured and taken from ref [32]. The interaction radius (*R*_0_) (Figure 13) is the extent of the solubility sphere encompassing the good solvents and excluding the bad ones. The Hansen parameters for a copolymer, shown in Table 6, were calculated based on the weight fraction (*w*_1_ and *w*_2_) of each monomer present in the copolymer chain and individual homopolymer Hansen parameters [33,34] using Equation (10). The interaction radius (*R*_0_) of the copolymer was also calculated as the weight average of individual homopolymers. The calculated Hansen parameters and interaction radius for Polymers 1–6 are shown in Table 6.
(10)$$\delta_{block-copolymer}=w_{1}\delta_{1}+w_{2}\delta_{2}$$

For a polymer, its solubility in a solvent or solvent blend depends on the Hansen solubility parameters of solvent being within the solubility sphere of the polymer (Figure 13). The distance (*D_S-P_*) of the solvent from the center of the solubility sphere can be calculated using Equation (11):(11)$$D_{S\text{-}P}=\left[4{\left(\delta_{ds}-\delta_{dp}\right)}^{2}+{\left(\delta_{ps}-\delta_{pp}\right)}^{2}+{\left(\delta_{hs}-\delta_{hp}\right)}^{2}\right]$$
where *δ_xs_* is the Hansen component parameter for the solvent composition THF/water at the collapse point, and *δ_xp_* is the Hansen component parameter for the polymer. If the distance (*D_S-P_*) is less than the interaction radius, the polymer is expected to dissolve in the solvent. A recent study [35] shows that the predictive accuracy of Hansen parameters is limited and is found to be 67% for solvents and 76% for non-solvents.

The distance (*D_S-P_*) calculated using Equation (11) for the water/THF solvent composition at the collapse point for Polymers 1–6 is shown in Table 6. The distance is higher than the interaction radius for all the copolymers. This indicates that the solvent mixture is a non-solvent for the polymer, which results in its collapse. The interaction radius of the copolymers gives a close, but not true, picture of the solubility behavior. This is due to the Hansen parameters and the interaction radius of the copolymer calculated using polymethacrylic acid. However, the copolymer, during the water-reduction process, is ionized, and the carboxylic acid groups are present as sodium salt or triethyl amine quaternary salt. However, the Hansen parameters for polymethacrylic acid sodium salt, or any other polyelectrolyte-type polymers or copolymers, have not been reported in any publications. The total Hansen parameter (*δ_T_*) for the water/THF solvent composition at the collapse point for all the polymers is shown in Table 4. For polymers with a low charge density (Polymer 4), the *δ_T_* value of the solvent composition is lower than that of a polymer with high charge density (Polymer 3). Increasing the hydrophobicity in the polymer chain, therefore, affects the Hansen parameters required for the solvent composition, which, in this case, becomes lower. Similar observations can be seen in Polymers 5 and 6, which were made using different monomers, as well as in the case of Polymer 2* when it is neutralized using a hydrophobic base such as triethylamine. The major contribution for the variation in the *δ_T_* of different polymer structures is due to the hydrogen bonding (*δ_h_*) component of the Hansen parameter as compared to the polar (*δ_p_*) component. The dispersive component shows the least variation due to changes in the hydrophobic–hydrophilic balance of the polymer. This observed trend is not surprising, as adding more ionic groups in the polymer chain will increase the affinity of the polymer towards the aqueous system, since the carboxylate anions on the polymer chain can hydrogen bond with water. When comparing Polymers 3 and 4 against Polymers 5 and 6, we can see the effect of using different types of hydrophobic monomers on the Hansen parameters. For a low charge density (high hydrophobicity), Polymer 4 (made using MMA and MAA monomers) has *δ_T_* = 33.15 at the collapse composition, whereas Polymer 5 (made using BMA, EMA and MAA) has *δ_T_* = 32.97, which is lower. This is also observed in high-charge-density polymers made using the same monomers (Polymers 3 and 6). The small difference due to the incorporation of a butyl methacrylate monomer can be attributed to the different Hansen parameters of PnBMA and PMMA, shown in Table 5. PnBMA has a lower *δ_T_* value than PMMA.

## 4. Conclusions

The AFM images of the CUP particles show the spheroidal conformation of the particle and verify the diameter of the particle observed in DLS. The collapse composition of the water/THF mixture was successfully determined using viscosity measurements. The use of a vibration viscometer for continuous viscosity measurements during the addition of water made it possible to obtain reproducible collapse points with less effort and better accuracy. The vibration viscometer was an ideal tool for this study, since it provided stable viscosity values with minimum noise, even when the solution was under constant stirring. The viscosity of the CUP polymer shows a steady rise in viscosity with the addition of water until it reaches the collapse composition. The rise in viscosity overcomes the dilution effect caused by the addition of water. After reaching the collapse composition, the viscosity drops, which is due to the polymer transforming from an extended coil into a spheroidal particle. The composition of the water/THF mixture at collapse changes as the co-polymer structure is varied. The collapse point ranges from a 53.8 to 59.3% water volume in the water/THF solvent composition, depending on the co-polymer structure. Adding more hydrophobicity to the copolymer reduces the amount of water required to trigger the collapse, as seen from Polymers 4 and 5. The dielectric of the solvent mixture plays an important role in separating the ion pair so that charges are felt over a longer distance. The dielectric at the collapse ranges from 45.5 to 49.6; however, the dielectric required for strong ion–ion repulsion to provide a stable particle must be below the dielectric at the collapse point. Altering the copolymer structure changes its Hansen solubility parameter. This changes the composition of the solvent mix where it is a poor solvent for the polymer, thereby leading to collapse. 

## Figures and Tables

**Figure 1 polymers-14-01909-f001:**
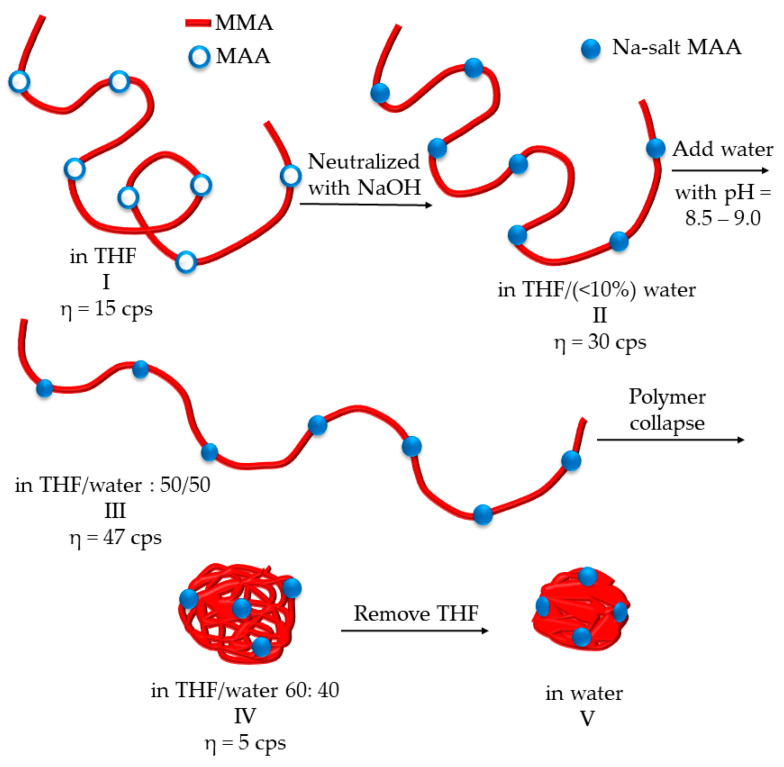
Schematics of the water-reduction process and CUP formation.

**Figure 2 polymers-14-01909-f002:**
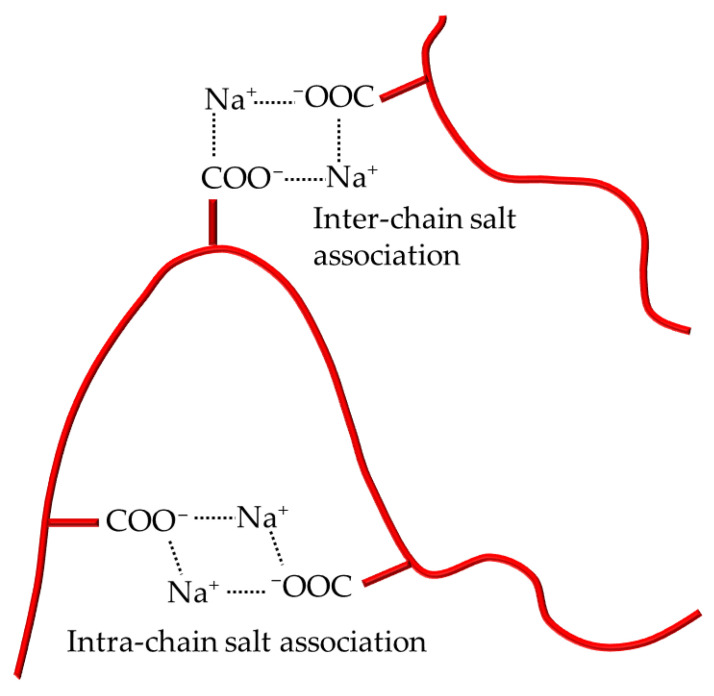
Inter- and intra-chain salt associations in polymer.

**Figure 3 polymers-14-01909-f003:**
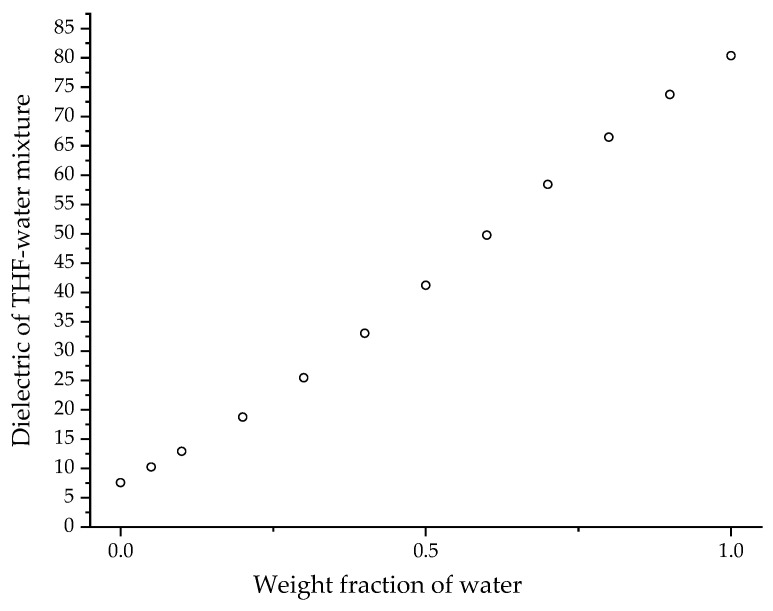
Dielectric of water/THF mixture. Data taken from [7].

**Figure 4 polymers-14-01909-f004:**
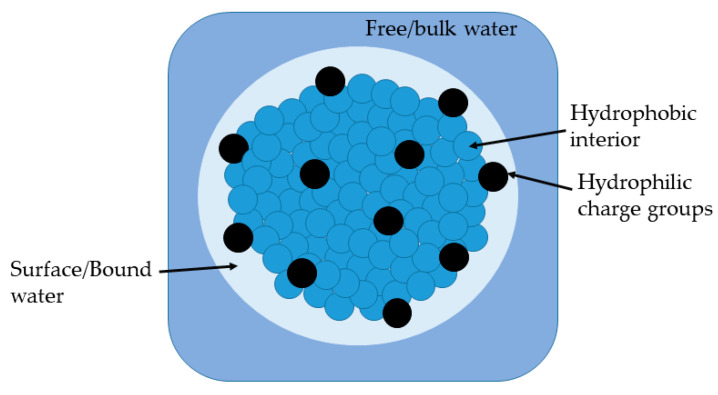
Structure of CUP particle suspended in water.

**Figure 5 polymers-14-01909-f005:**
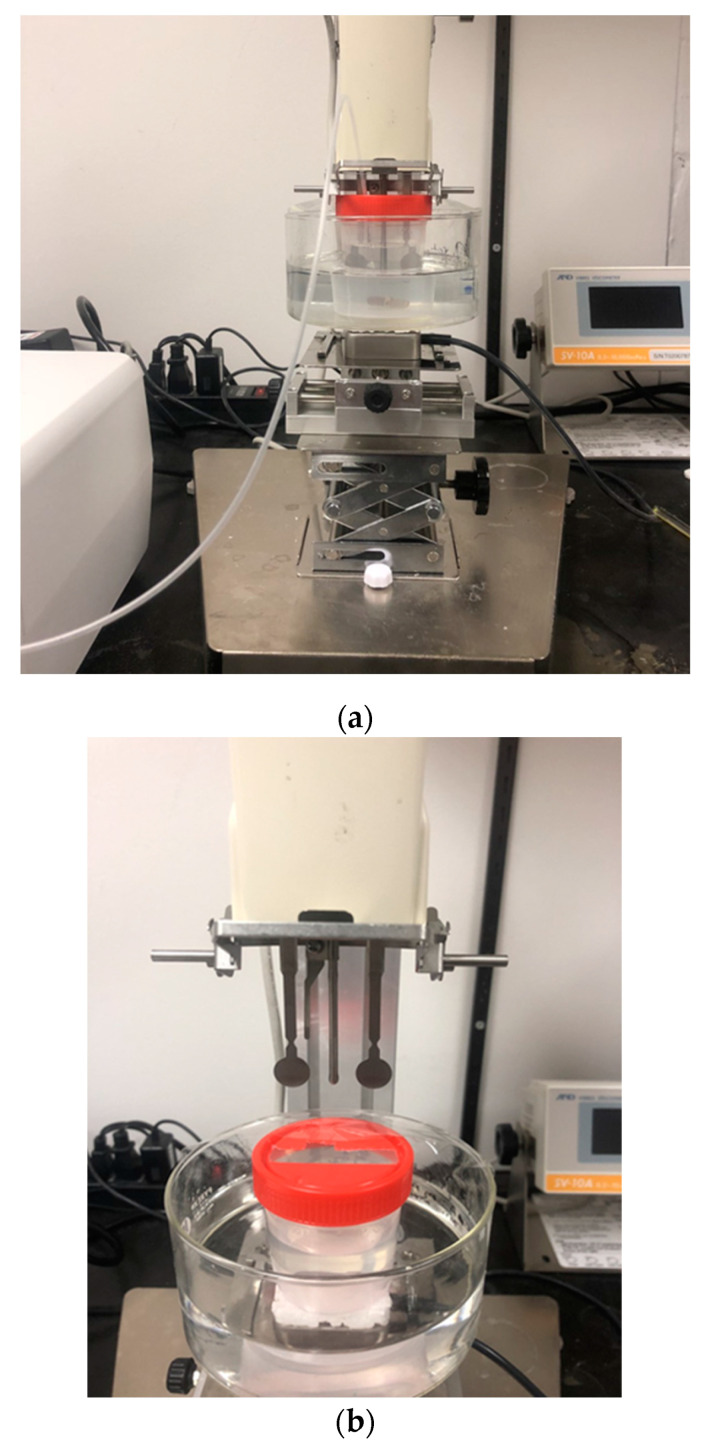
(**a**) Experimental set-up used for testing the continuous process for collapse-point determination and for measuring the collapse point of CUP polymer; (**b**) picture of tuning forks on the vibration viscometer.

**Figure 6 polymers-14-01909-f006:**
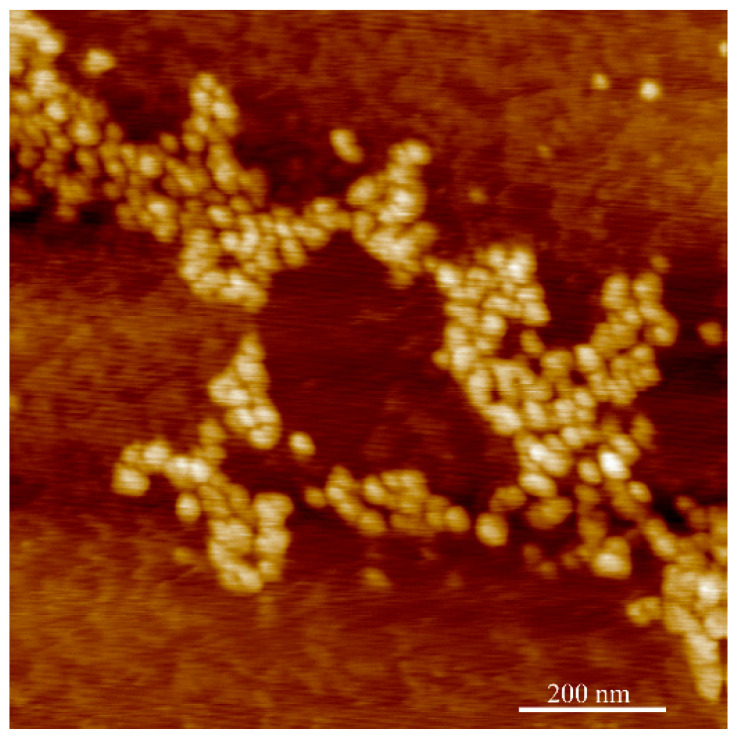
AFM image showing Polymer 2 CUP particles in dense clusters or aggregations.

**Figure 7 polymers-14-01909-f007:**
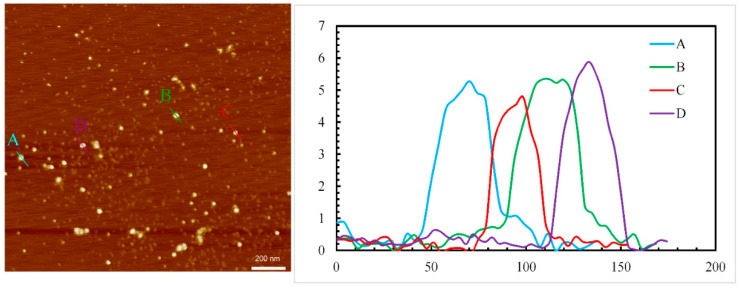
AFM image showing sparsely clustered or aggregated Polymer 2 CUP particles (left) and the height (nm—y axis) and width (nm—x axis) of the profile of the analyzed particles (A–D).

**Figure 8 polymers-14-01909-f008:**
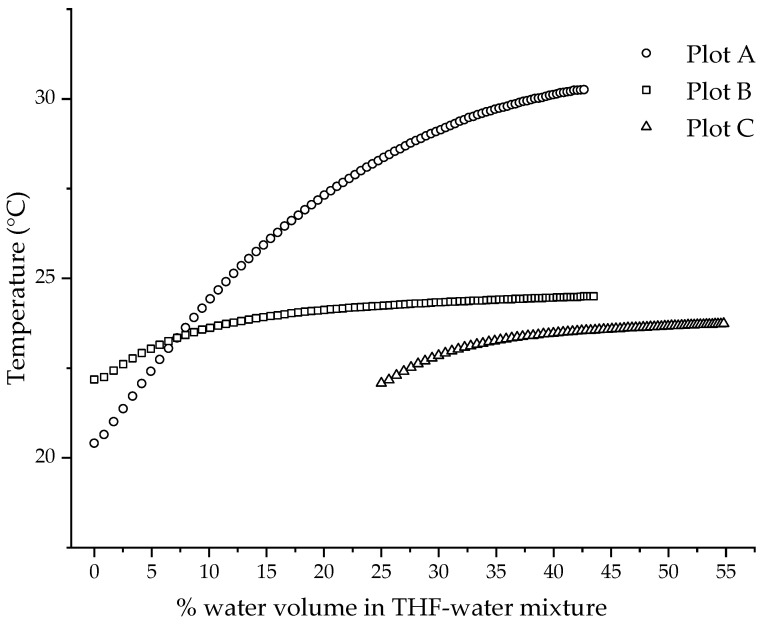
Temperature profile for the addition of water obtained by making three different changes to experimental set-up.

**Figure 9 polymers-14-01909-f009:**
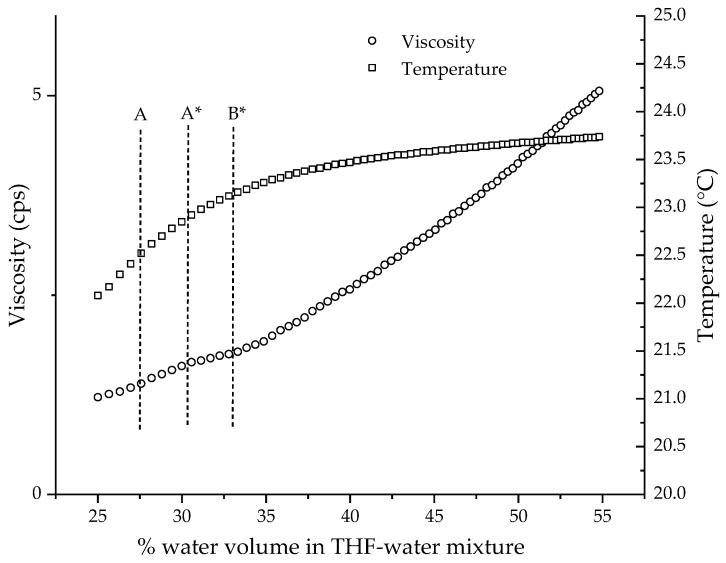
Viscosity against time for THF/water mixture of 75/25 *v*/*v* composition with addition of water at different levels (A, A*, B*) on tuning fork shown in Figure 10.

**Figure 10 polymers-14-01909-f010:**
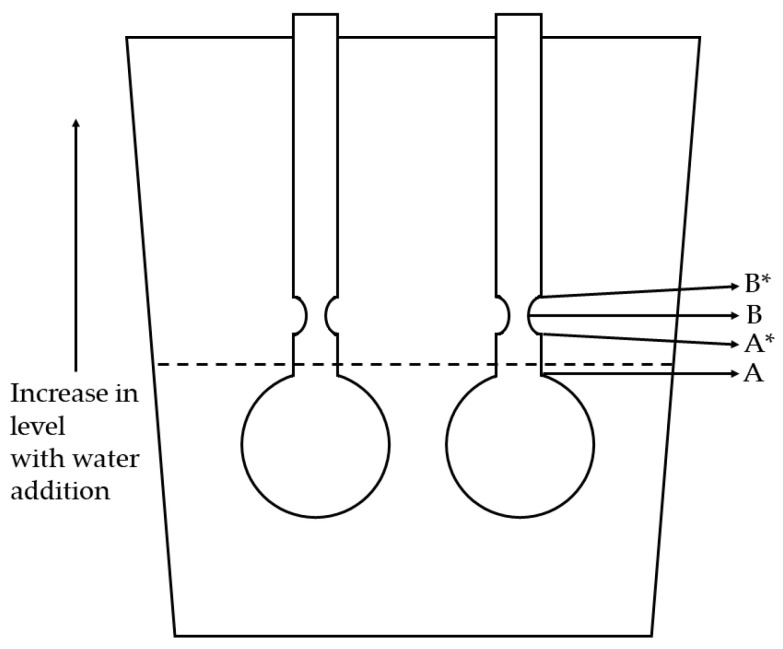
Depiction of solution levels and immersion of tuning forks.

**Figure 11 polymers-14-01909-f011:**
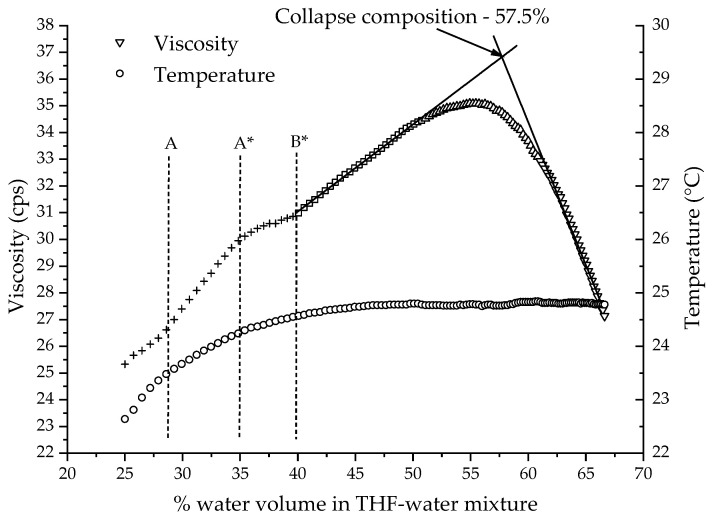
Water reduction (viscosity against % volume of water present in the THF/water solvent composition) plot for Polymer 2.

**Figure 12 polymers-14-01909-f012:**
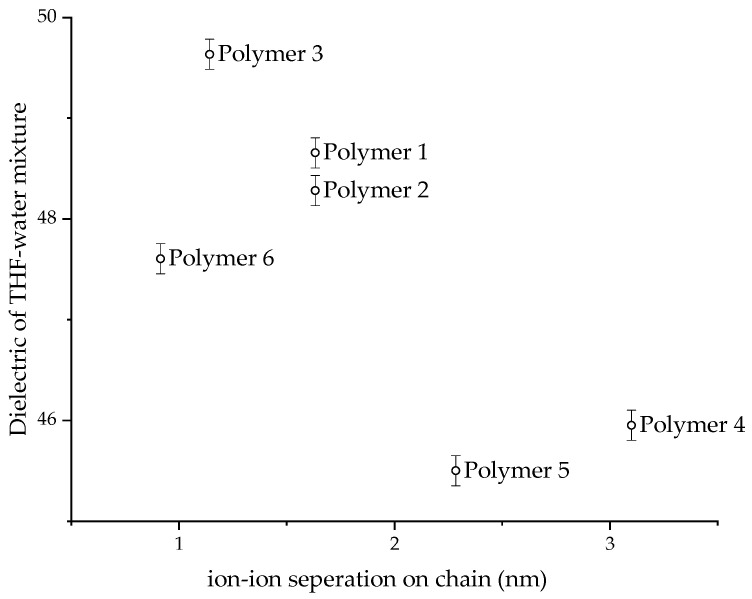
The dielectric of THF/water mixture at collapse point against ion–ion separation on the polymer chain.

**Figure 13 polymers-14-01909-f013:**
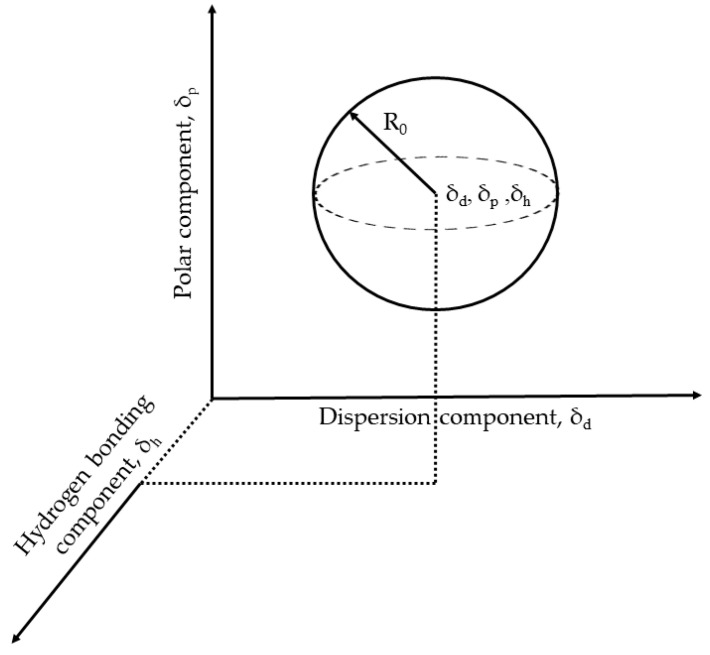
The Hansen volume of solubility for a polymer is depicted using a 3-D model of solubility sphere with center at (*δ_d_*, *δ_p_*, *δ_h_*) and radius of interaction (*R*_0_). Liquids whose parameters lie within the volume of the solubility sphere are active solvents for that polymer. Reprinted (adapted) with permission from Venkatram, S.; Kim, C.; Chandrasekaran, A.; Ramprasad, R. Critical assessment of the Hildebrand and Hansen solubility parameters for polymers. J. Chem. Inf. Model. 2019, 59, 4188–4194. Copyright 2019 American Chemical Society.

**Table 1 polymers-14-01909-t001:** Molar quantities of monomers, initiator (AIBN) and chain transfer agent (1-dodecanethiol) used for synthesis of Polymers 1–6.

Polymer	MMA (mol)		MAA (mol)	AIBN (mol)	1-Dodecanethiol (mol)	THF (mol)
1 ^a^	0.912		0.101	7.09 × 10^−4^	3.49 × 10^−3^	2.77
2 ^a^	0.912		0.101	7.09 × 10^−4^	1.45 × 10^−3^	2.77
3	0.874		0.146	7.14 × 10^−4^	1.46 × 10^−3^	2.77
4 ^a^	0.953		0.053	7.04 × 10^−4^	1.60 × 10^−3^	2.77
	**EMA (mol)**	**BMA (mol)**	**MAA (mol)**			
5	0.279	0.446	0.056	5.46 × 10^−4^	1.27 × 10^−3^	2.77
6	0.330	0.413	0.040	5.49 × 10^−4^	1.28 × 10^−3^	2.77

^a^ All the data for Polymers 1, 2 and 4 are taken from Ref [12].

**Table 2 polymers-14-01909-t002:** Acid number, densities, molecular weight and polydispersity index (PDI) of the copolymers.

Sample ID	*M_W_* (g/mol) ^b^	PDI	Monomer Ratio (MMA:MAA)	AN Meas./calc. (mg KOH/g) ^c^	Density Dry, *ρ_p_* (g/mL)
Polymer 1 ^a^	28.9 K	1.8	9:1	56.8/56.9	1.2246 ± 0.0018
Polymer 2 ^a^	59.8 K	1.7	9:1	57.0/56.9	1.2311 ± 0.0014
Polymer 3	33.0 K	1.9	6:1	81.3/81.7	1.2300 ± 0.0012
Polymer 4 ^a^	45.4 K	1.8	18:1	29.1/29.7	1.2390 ± 0.0019
			**(EMA:BMA:MAA)**		
Polymer 5	51.1 K	1.7	5:8:1	31.5/31.3	1.2370 ± 0.0021
Polymer 6	47.1 K	1.7	1:3.6:1	78.5/78.8	1.2353 ± 0.0017

^a^ All the data for Polymers 1, 2 and 4 are taken from Ref [12]; ^b^ absolute number average molecular weight from GPC; ^c^ AN—acid number, measured using ASTM D974.

**Table 3 polymers-14-01909-t003:** Measured and calculated particle size (diameter), polydispersity index of particle size (PS-PDI), width at half height (WHH) and charge density of the CUPs.

Sample ID	d(DLS) ^b^ (nm)/PDI ^c^	WHH ^d^ (nm)	d(GPC) ^e^ (nm)	Charge Density, *ρ_v_*, (Ions per nm^2^)
Polymer 1 ^a^	4.22/1.02	0.61	4.25	0.52
Polymer 2 ^a^	5.38/1.02	0.71	5.40	0.66
Polymer 3	4.38/1.01	0.49	4.42	0.80
Polymer 4 ^a^	4.90/1.01	0.41	4.92	0.32
Polymer 5	5.12/1.02	0.58	5.10	0.35
Polymer 6	5.00/1.02	0.65	4.97	0.84

^a^ All the data for Polymers 1, 2 and 4 are taken from ref [12]; ^b^ diameters measured using dynamic light scattering (DLS) instrument; ^c^ PS-PDI = d_w_/d_n_; d_w_ = ∑N_i_d_i_/∑N_i_; d_w_ = ∑N_i_d_i_^2^/∑N_i_. N and d = number (% channel) and diameter of particles at “ith” intensity. Due to the low resolution of DLS instrument, PDI is much smaller than that observed in the corresponding co-polymers; ^d^ WHH—width at half height of the particle size distribution curve; ^e^ diameters calculated from average absolute molecular weight measured using gel permeation chromatography (GPC) using Equation (1).

**Table 4 polymers-14-01909-t004:** Collapse composition for polymers 1-6 and Hansen parameters of solvent composition at collapse.

Sample ID	Volume Composition at Collapse Point		*δ_d_*	*δ_p_*	*δ_h_*	*δ_T_*	Dielectric
	% Water	% THF					
Polymer 1	58.0	42.0	16.1	11.7	27.9	34.3	48.7
Polymer 2	57.5	42.5	16.1	11.6	27.7	34.1	48.3
Polymer 3	59.3	40.7	16.1	11.8	28.3	34.7	49.6
Polymer 4	54.4	45.6	16.2	11.3	26.7	33.2	45.9
Polymer 5	53.8	46.2	16.2	11.2	26.5	33.0	45.5
Polymer 6	56.6	43.4	16.1	11.5	27.4	33.8	47.6
Polymer 2 *	56.2	43.8	16.1	11.5	27.3	33.7	47.3

* Polymer 2 was neutralized using triethyl amine instead of sodium hydroxide.

**Table 5 polymers-14-01909-t005:** Hansen parameters and the interaction radius (*R*_0_) for the homopolymers. Data taken from [32].

Homopolymer	*δ_d_*	*δ_p_*	*δ_h_*	*δ_T_*	Interaction Radius, *R*_0_
Poly methyl methacrylate (PMMA)	19.1	11.3	4.1	22.6	10.3
Poly ethyl methacrylate (PEMA)	19.0	9.0	8.0	22.5	11.0
Poly n-Butyl methacrylate PnBMA	16.0	6.2	6.6	18.4	9.5
Poly methacrylic acid (PMAA)	25.6	11.2	19.6	34.1	20.3

**Table 6 polymers-14-01909-t006:** Hansen parameters, interaction radius (*R*_0_) and the distance (*D_S-P_*) at collapse composition for Polymers 1–6.

Co-Polymer	*δ_d_*	*δ_p_*	*δ_h_*	*δ_T_*	Interaction Radius, *R*_0_	Distance *D_S-P_*
Polymer 1	19.7	11.3	5.5	23.3	11.2	23.4
Polymer 2	19.7	11.3	5.5	23.3	11.2	23.6
Polymer 3	19.9	11.3	6.0	23.7	11.6	23.6
Polymer 4	19.4	11.3	4.8	23.0	10.8	22.8
Polymer 5	17.4	7.3	7.7	20.4	10.5	19.4
Polymer 6	17.6	7.3	8.4	20.8	11.0	19.7

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
