# Peer review of "Defining the Collapse Point in Colloidal Unimolecular Polymer (CUP) Formation"

_polymers, 2022, doi:10.3390/polym14091909_

Round 1

Reviewer 1 Report

Dear authors,

Here are my comments for this paper:

  1. Show the source for monomers and initiator which are not purified (company).
  2. The reference for density values should be shown.
  3. The number of references is quite small. You have to add more references to support the introduction and the results/discussion sections.
  4. The novelty and added value of the manuscript must be added (with respect to existent literature).
  5. Which is the target application/applications for these CUP particles?
  6. PDI values in DLS analysis.
  7. The paper can benefit from some morphological characterization of the particles (TEM).

Author Response

We are thankful for your time in reviewing “Polymers – 1681019” manuscript and providing your crucial insights on this research. We have addressed all the revisions and questions in the order they were mentioned in the review report.  

  1. The source for initiator (AIBN) and chain transfer agent (1- dodecanethiol) has been mentioned in the section 2.1 – Materials, synthesis, and characterization in line 175-176. They were sourced from Sigma Aldrich.
  2. The detail procedure and methods for characterizing the polymer which includes the molecular weight, acid number and dry polymer density and characterizing the CUP particle size were taken from and have been rightfully cited to reference no. 12 – “Geng, P.; Zore, A.; Van De Mark, M.R. Thermodynamic Characterization of Free and Surface Water of Colloidal Unimolecular Polymer (CUP) Particles Utilizing DSC. Polymers 2020, 12, 1417”. All the characterization results shown for polymers 1, 2 and 4 (as denoted using superscript “a”) in Table 2 are taken from reference no. 12 whereas characterization for the remaining polymers was done by us using the procedure and methods mentioned in reference no. 12. The source of procedure and methods for characterizing of polymers and CUP particle have been addressed in the paper using additional statements (line 167-170) and proper citation. The title of the section 2.1 has been changed from “Materials and synthesis” to “Materials, synthesis, and characterization” which will clarify that the source of methods and procedure for characterization is mentioned in this section.
  3. 9 Additional references (see reference section) pertaining to research in single chain nanoparticles has been included in the manuscript.
  4. The set up developed in this study allowed, for the first time, rapid and precise determination of the collapse point where the polymer chain transforms into a collapsed particle. This enabled us to change the polymer structure and composition and observe the effect on the collapse point i.e., where the chain transforms into a particle. The technique developed in this study can potentially be used to study Flory Huggins collapse behavior in other polymeric systems. This novelty and added value have been addressed in line 155-163.
  5. The target applications of these CUP particles were discussed earlier. The discussion on the application have been further expanded in line 77-82 to include some more details and specifics.
  6. The PDI values for DLS has been mentioned in the Table 3.
  7. The CUP particles have been observed under TEM. However, due to aggregation of these particles after drying, it shows clusters instead of individual particles thereby giving poor resolution.

Reviewer 2 Report

Comments

  1. The introduction part is very exhaustive and too long, it should be resumed.
  2. What the star marks indicating in Table 1?
  3. Line 230-231: The polymer stock solution was 230 prepared by dissolving the polymer in THF to make a 15% w/w solution? Why w/w, why not w/v?
  4. The dot plots as shown in Figure 10, should be with standard error bar.
  5. The methods for the sections 3.2. Particle size analysis and 3. Charge density of the CUP particle should be mentioned in the materials and method section rather than mixing the methodology and results together in the results and discussion section.
  6. Out of 26 references in this manuscript, more than 10 are self-citation.
  7. The abstract and conclusions are just the summary or results no data or values were mentioned anywhere.
  8. Whole manuscript moves around vibration viscometer only.
  9. What was the purpose of Figure 8, Depiction of solution levels and immersion of tuning forks? Is the manuscript being a review article or a book chapter or detail procedure about the handling of vibration viscometer?
  10. Most of the experiments were performed in triplicate as mentioned by the authors, but I do not find any figure with standard error bars.

Author Response

We are thankful for your time in reviewing “Polymers – 1681019” manuscript and providing your crucial insights on this research. We have addressed all the revisions and questions in the order they were mentioned in the review report. 

  1. The introduction covers critical information such the theory for conformation changes in polymer, introduction to our CUP system, applications of CUP system, previous work done with conformation changes in polymer and single chain nanoparticles and its significance with respect to the current work and the aim and novelty of the current research. The information provided in the introduction is necessary to provide the foundation for the research that follows. As per the comments and suggestion from the other reviewer, the introduction was asked to be expanded on certain aspects.
  2. The star marks in Table 1 are the multiplication signs. They have been changed to the following “×” multiplication symbol.
  3. The solutions were prepared based on weight because of the accuracy of weight measurements over volume since, volume is temperature dependent. The weights were later converted to volume because the Hansen parameters of the mixture of water and THF are calculated (equation 7-9) using volume fractions.
  4. The dot plots in Figure 10 have been shown with error bars.
  5. The section 3.2. Particle size analysis - is the discussion on the measured particle size with reference to our previous work and work done with proteins. It also shows the relation between the size and molecular weight. The lines 256-258 in section 3.2 have been removed to avoid misleading interpretation of the result and discussion that follows. The methodology and procedure for particle size measurement has been published in reference no.12. In this manuscript, size measurement procedure will be mentioned as a link to reference no. 12 in section 2.1 - “Materials, synthesis, and characterization”. The charge density of the CUP particle is calculated parameter using equation 2. Since it is not a measured parameter, it was mentioned in the results and discussion section.
  6. 9 more references have been added to relate other work done in this area.
  7. The abstract (line 17-19) and summary (line 634-635, 639-641) includes some critical experimental data.
  8. The goal of this research was to find an easier, more rapid method for the determination of collapse point. This required us to be able to continuously measure the viscosity during water addition. The process also required mixing while water is added. Other viscometers like cone and plate, Ubelhode for example cannot be used for continuous measurements. Vibration viscometer was the only one that allows continuous measurement while stirring without affecting the measurement stability (also discussed in detail in line 389-403). This is also addressed in the introduction section in line 153-163.
  9. During the water addition experiment, the level of the solution is continuously increasing. This level increases immerses the tuning fork further into the solution which in turn affects the viscosity values. Figure 7 and figure 8 both shows the effect of increase in solution level on the viscosity values where viscosity changes slope due to the change shape of tuning fork. This has been discussed in detail in line 404-432.
  10. Error bars have been added to the figures 10. For figure 7, 8 and 9 the error in the viscosity and temperature is same as the accuracy of the instrument mentioned in the section 2.2 – viscosity measurements.

We hope you find all the revision done to the manuscript satisfactory.

Sincerely,

Round 2

Reviewer 1 Report

Dear author,

The paper has been seriously improved and could be accepted after some minor revision. Please add some morphological characterization at least as supplimentary material (SEM, TEM). I understand the cluster formation, but still some morphological data are important.

Reviewer 2 Report

Authors have responded the raised comments. 

Author Response

Thank you for all your insightful and critical suggestions.